# Deep Spiking Neural Network with Brain-Inspired Recurrent Iterative Learning

## Abstract

Spiking neural networks (SNNs) have emerged as a transformative paradigm in artificial intelligence, offering event-driven computation and exceptional energy efficiency. However, conventional SNN training methods predominantly rely on backpropagation with surrogate gradients, often neglecting biologically plausible mechanisms such as spike-timing-dependent computations and dynamic excitation-inhibition balance—key features that underpin the brain's remarkable efficiency and adaptability. To bridge this gap, we propose Brain-Inspired Recurrent Iterative Learning (BIRIL), a novel hybrid learning framework that synergistically integrates biologically realistic spike transmission with adaptive excitation-inhibition dynamics. BIRIL not only emulates the temporal precision of biological neurons but also dynamically modulates neuronal activity to enhance learning efficiency. Extensive experiments on benchmark datasets—including CIFAR-10, CIFAR-100, MNIST, and DVS128 Gesture—demonstrate that BIRIL outperforms state-of-the-art SNN models, achieving superior accuracy while maintaining low computational overhead. Our work provides a principled approach to advancing neuromorphic learning, paving the way for more brain-like and energy-efficient AI systems.

## 1 Introduction

Artificial Neural Networks (ANNs) have driven transformative progress in machine learning, achieving state-of-the-art performance in domains such as computer vision and natural language processing. Yet, their reliance on high-precision floating-point computations and global error backpropagation Werbos (1990), diverges fundamentally from the brain's efficient, event-driven mechanisms. In contrast, Spiking Neural Networks (SNNs)—the third generation of neural networks Maass (1997)—offer a biologically grounded paradigm, combining event-driven sparsity with ultra-low energy consumption. These properties position SNNs as a compelling alternative to conventional ANNs for next-generation AI hardware. However, despite their neuromorphic potential, most deep SNN training methods remain tethered to surrogate gradient backpropagation Wu et al. (2018), mirroring ANN limitations. This approach not only overlooks the critical role of spike-timing-dependent computations but also fails to incorporate dynamic excitation-inhibition balance, a hallmark of biological learning. Consequently, existing SNNs underutilize their inherent spatiotemporal processing capabilities and face barriers in deploying adaptive on-chip learning for neuromorphic hardware. To bridge this gap, we propose a hybrid learning framework for deep SNNs that synergistically integrates local spike-driven plasticity with global error modulation. By co-optimizing biologically plausible dynamics and task-driven performance, our approach unlocks the full potential of SNNs for efficient, hardware-friendly neuromorphic systems.

Local and global learning paradigms offer complementary advantages for training SNNs. On one hand, local learning rules, such as spike-timing-dependent plasticity (STDP) Song et al. (2000), closely mimic biological synaptic plasticity by updating weights based on precise temporal relationships between pre- and post-synaptic spikes. These biologically inspired mechanisms enable fully local credit assignment, eliminating dependence on a global loss function while enhancing robustness and hardware compatibility for on-chip learning. On the other hand, global learning methods like spatio-temporal backpropagation (STBP) Wu et al. (2018) leverage surrogate gradients to achieve competitive performance in deep SNNs, albeit at the cost of biological plausibility. While these approaches differ fundamentally, their synergistic integration presents a promising av-

enue for developing efficient and versatile SNN training frameworks. Recent work has explored this hybrid approach, such as EICIL Shao et al. (2023), which dynamically combines STDP-based local learning with STBP-driven global updates using an excitation-inhibition gating mechanism. However, while EICIL demonstrates the feasibility of hybrid learning, its performance remains limited in deep SNN architectures, highlighting the need for more scalable and adaptive integration strategies.

Decades of neuroscientific research have revealed that biological neural networks operate through sophisticated synaptic mechanisms Raichle & Mintun (2006), where information is encoded and transmitted via dynamic electro-chemical signal conversion. A fundamental feature of this system is the balanced interplay between excitatory and inhibitory neurons Lauritzen et al. (2012), which modulates downstream neural activity through precisely regulated neurotransmitter release. This delicate excitation-inhibition (E-I) balance is crucial for maintaining stable yet adaptable network dynamics. Therefore, we consider alternating excitation and inhibition during the learning process and designan iterative hybrid global and local learning mechinism in deep SNNs.

In this paper, we propose a novel training framework named Brain-Inspired Recurrent Iterative Learning (BIRIL) for deep SNNs. BIRIL employs a three-cycle training strategy. The strategy comprises an excitation, an inhibition, and an excitation-inhibition mechanism. By mixing these three mechanisms in different proportions, the distribution of excitatory and inhibitory training iterations can be dynamically rebalanced. Hence BIRIL can realistically simulate the excitation/inhibition state of biological neurons, and integrate a parameter update mechanism for both the time and spatial dimension during training. Thus, the BIRIL training framework has better biological interpretability and resembles the real working structure of the human brain.

The contributions of this paper are summarized as follows:

- This paper proposes a novel hybrid training framework based on BIRIL, which enhances the performance of deep SNNs trained with hybrid learning.

- We propose a novel three-cycle training strategy that integrates excitation, inhibition, and the excitation/inhibition mechanism, enhancing the spatiotemporal dynamics of SNNs through the dynamic adjustment of excitatory and inhibitory training iterations.

- Experimental results demonstrate that BIRIL significantly outperforms locally/globally learned SNNs onmultiple benchmark datasets. The results show that theappropriate distribution ratio of BIRIL is quite close to the excitation-inhibition property in the human brain.

## 2 RELATED WORK

### 2.1 LEARNING METHODS OF SNNS

Spiking Neural Networks (SNNs) employ two principal learning paradigms: unsupervised local learning and supervised global learning. The first category is exemplified by Spike-Timing-Dependent Plasticity (STDP), a biologically inspired mechanism that adjusts synaptic weights based on temporal correlations between pre- and postsynaptic spikes Song et al. (2000)Nessler et al. (2009). STDP strengthens connections when presynaptic neurons fire before postsynaptic neurons, while weakening them in the reverse temporal order. Recent extensions, such as reward-modulated STDP (R-STDP), incorporate global neuromodulatory signals to enhance learning efficiency Mozafari et al. (2018). However, these biologically plausible approaches often face computational limitations due to their local nature and increased energy demands from reward-processing mechanisms.

The second category addresses supervised learning through adaptations of backpropagation for spiking networks. While conventional backpropagation Werbos (1990) is incompatible with SNNs' discrete spiking dynamics, the Spatio-Temporal Backpropagation (STBP) algorithm overcomes this by employing surrogate gradients to enable differentiable training Wu et al. (2018). STBP effectively captures both spatial and temporal information flow, but its reliance on global error signals and computationally intensive backpropagation through time presents scalability challenges. This fundamental trade-off between biological plausibility and computational efficiency remains a key consideration in SNN learning algorithm design.

## 2.2 BRAIN-INSPIRED HYBRID TRAINING

Unlike purely local or global training methods for Spiking Neural Networks (SNNs), hybrid approaches that integrate global Spatio-Temporal Backpropagation (STBP) with local Spike-Timing-Dependent Plasticity (STDP) Yan et al. (2021) provide a more biologically plausible framework by better emulating the spiking dynamics of biological neurons. These methods leverage both STBP and STDP to update synaptic weights, combining the strengths of global gradient-based optimization with local, biologically inspired plasticity rules. For instance, the EICIL method Shao et al. (2023) alternates between STDP and STBP during training, capturing excitatory and inhibitory neuronal behaviors to enhance learning. However, EICIL's sequential application of STDP and STBP—rather than concurrent optimization—limits its effectiveness, often yielding suboptimal performance. Another approach, Hybrid Plasticity (HP) Wu et al. (2022), separately updates parameters using STBP and STDP before integrating them into the final model. While HP improves accuracy, its dual-parameter optimization introduces significant computational overhead. To address these limitations, we propose an efficient hybrid training method that synergistically combines STDP and STBP, enhancing SNN performance while more faithfully replicating the brain's biological learning mechanisms.

## 3 PRELIMINARY

### 3.1 INTEGRATE-AND-FIRE NEURON MODEL

Integrate-and-Fire (IF) Gerstner et al. (2014) is an important module that simulates the activity of biological neurons in SNNs. IF neurons receive pulse inputs from other neurons and send output pulses to other neurons based on the accumulated charges. As a discrete-time model, IF neurons accumulate each input pulse to generate the membrane potential. If the membrane potential reaches the firing threshold, the neuron outputs a pulse and resets the membrane potential.

At each time step, the membrane potential of an IF neuron accumulates membrane charges from other neurons. The calculation formula of the IF neuron is as follows:

$$V(t) = V(t-1) + \sum_i I_i(t), \tag{1}$$

where $V(t)$ represents the membrane potential at the current time step, $V(t-1)$ represents the membrane potential at the previous time step, and $\sum_i I_i(t)$ represents the input current from the other neuron.

When the membrane potential exceeds a threshold $V_{\text{th}}$, the IF neuron fires a pulse:

$$S(t) = \begin{cases} 1, \ if \ V(t) \ \geq V_{\text{th}}, \\ 0, \ if \ V(t) \ < V_{\text{th}}, \end{cases} \tag{2}$$

where $S(t)$ represents the spike train, denoting whether the neuron releases a spike at the current time step, and $V_{\text{th}}$ is the threshold that triggers the spike release. If the IF neuron fires a spike, it resets the membrane potential.

### 3.2 NEURONAL EXCITATION AND INHIBITION

Neurons in the human brain are connected by synapses, which transmit impulse signals mainly by releasing neurotransmitters. Excitatory neurons and inhibitory neurons produce different effects by releasing different neurotransmitters. The two types of neurons form complex and sophisticated local circuits that play a vital role in regulating the higher brain functions of the cerebral cortex. In different areas of the human brain, the number of excitatory neurons and inhibitory neurons has different ratios. For example, the ratio of excitatory neurons to inhibitory neurons in the cerebral cortex is roughly 4:1. Excitatory neurons promote the strength of synapses through mechanisms such as long-term potentiation (LTP) to strengthen memory. Inhibitory neurons prevent excitatory neurons from overactivity and reduce interference between new information and existing memories,

making memory retrieval more explicit. The interaction between neuronal excitatory and inhibitory properties plays a vital role in the ability of the human brain's nervous system to process information. Motivated by the above, we think that the training process of SNNs could be adjusted according to the inhibitory and excitatory distribution.

### 3.3 SPIKE-TIMING-DEPENDENT PLASTICITY(STDP)

STDP is widely believed to be a fundamental way of learning and information storage in the human brain. STDP is attractive because of its excellent biological interpretability. In the human brain, action potentials often change very precisely according to external stimuli, and STDP simulates this process very well. Specifically, STDP adjusts the strength of the connection based on the relative timing of the action potential input and output of a particular neuron. If an input to a neuron occurs before the output of the neuron, the synapse becomes stronger and becomes a Long-Term Potentiation (LTP) synapseMalenka & Nicoll (1999), while if an input occurs after the output of the neuron, the synapse becomes weaker and becomes a Long-Term Depression (LTD) synapseIto (1989).

The following equation formulates this process:

$$\Delta\omega = \begin{cases} A_+ \cdot e^{-(t_{\text{post}} - t_{\text{pre}})/\tau_+}, & if\ t_{\text{post}} > t_{\text{pre}}, \\ A_- \cdot e^{-(t_{\text{pre}} - t_{\text{post}})/\tau_-}, & if\ t_{\text{pre}} > t_{\text{post}}, \end{cases} \tag{3}$$

where $\Delta\omega$ represents the change in synaptic weight, positive values indicate an increase in weight, and negative values indicate a decrease in weight, $t_{\text{pre}}$ and $t_{\text{post}}$ represent the time when the pre-neuron and post-neuron fire pulses, respectively, $A_+$ and $A_-$ represent the amplitude of the increase and decrease in the regulating weight, and changing them can change the strength of synaptic plasticity changes, $\tau_+$ and $\tau_-$ are the time constants of weight enhancement and weakening, respectively, which determine the decay rate of synaptic weight changes.

This means that the smaller the time difference between the pulses, the more "important" the synapse is, and the greater the change in weight is. The larger the time difference between the previous and next pulses, the more "unimportant" the synapse is, and the smaller the change in weight is. If the time difference between the two pulses is large enough, the weight of the synapse may not change significantly.

### 3.4 SPATIO-TEMPORAL BACKPROPAGATION(STBP)

Spatio-Temporal Backpropagation is a method that combines the powerful capabilities of the traditional Backpropagation (BP) algorithm with good biological interpretability.

In back propagation, the loss function is formulated as follows:

$$L = \frac{1}{2S} \sum_{s=1}^{S} ||y_s - \frac{1}{T} \sum_{t=1}^{T} o_s^{t,N}||_2^2, \tag{4}$$

where $L$ is a function of weights and biases, where $S$ is the number of samples in this batch, $y_s$ is the sample label, $o_s^{t,N}$ is the output at time step $t$, and the gradient propagates along the time domain and space domain respectively.

Since the discrete spikes in SNNs result in non-differentiable gradients during backpropagation, alternative gradient functions are required to approximate continuous gradients from the discrete spike events, known as surrogate gradient methods for SNNs.

Among them, the derivative of the sigmoid function is a common and effective gradient substitute function:

$$sigmoid(x) = \frac{1}{1 + e^{-x}}. \tag{5}$$

When solving the derivative of the sigmoid function, we scale it by factor $\alpha$ to make the gradient calculated during backpropagation larger or smaller, thereby effectively accelerating the convergence process:

$$grad(x) = \alpha \cdot sigmoid(x) \cdot (1 - sigmoid(x)). \tag{6}$$

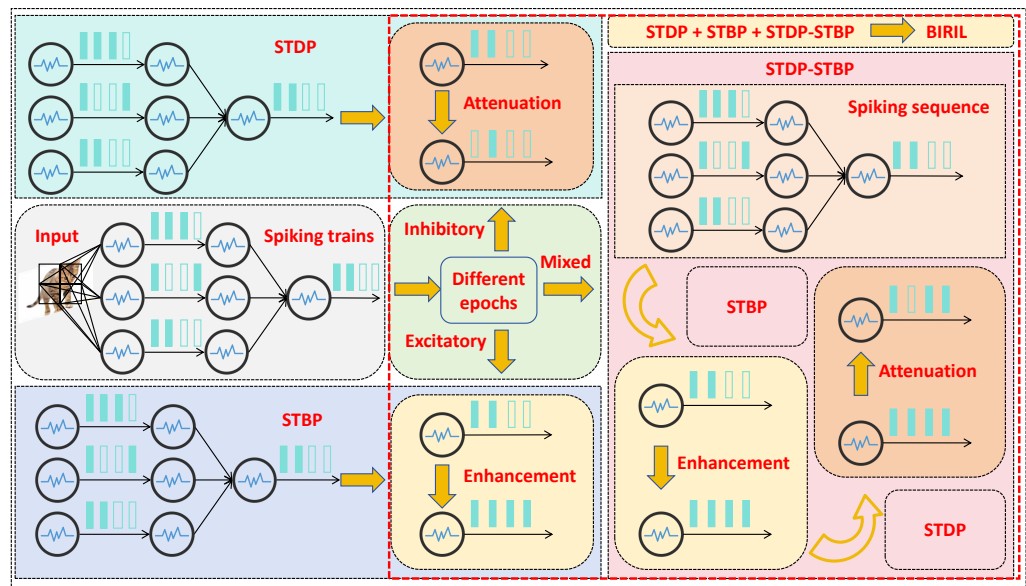

Figure 1: The figure above shows the training model framework, including the backbone network and the residual network, which ensures that the excitation mechanism and the inhibition mechanism can give full play to their respective advantages during training.

## 4 METHODS

### 4.1 MULTI-LAYER STDP

Due to challenges such as the high computational cost, the nature of neuronal activity, and the complexity of synaptic plasticity mechanisms, traditional STDP is typically applied to a single layer within a neural network, limiting the scope of parameter updates to that specific layer. However, with the advancement of research, multi-layer STDP Vignoud et al. (2018) has gained increasing attention as a means to extend the applicability of STDP to deeper network architectures.

To bridge this gap, we propose a hybrid STDP-STBP framework that integrates the biological plausibility of STDP with the global optimization capabilities of STBP. In this framework, STBP is used to update local synaptic weights, while STDP leverages the gradient information from STBP to perform backpropagation. This cyclic gradient update mechanism ensures effective error propagation to earlier layers, addressing the limitations of pure STDP in deep networks. Additionally, we incorporate biologically inspired mechanisms, such as synaptic normalization and spike gating, to further enhance the robustness and efficiency of deep SNN training. By combining the strengths of STDP and STBP, our approach not only preserves the biological principles of human brain learning and feedback but also achieves state-of-the-art performance in deep SNNs without reducing network depth.

### 4.2 STDP-STBP GRADIENT UPDATE MECHANISM

In the learning and memory mechanism of the human brain, there is not only unidirectional pulse transmission but also reverse pulse transmission. The pulse transmission between the neural networks in the human brain is a forward-reverse cycle mode. Previous studies tended to use only one STDP or STBP for unidirectional pulse transmission, which obviously does not conform to the information processing mechanism of the human brain. Therefore, we combined the characteristics of excitatory neurons and inhibitory neurons and the mechanism of neuronal pulse transmission in the human brain and used STDP and STBP to update the weights of neuronal connections. Among them, STDP is used to represent the inhibitory state of neurons, and STBP is used to represent the excitatory state of neurons. Using the different characteristics of STDP and STBP to update the

---

**Algorithm 1** STDP-STBP cyclic gradient update mechanism

---

**Require:** The connection weights between neurons $w$, Weights updated by STDP $\Delta w_{stdp}$, Weights updated by STBP $\Delta w_{stbp}$,The time when the front neuron fires a pulse $t_{pre}$,The time when the rear neuron fires a pulse $t_{post}$.

**for** $i = 1$ **to** $epoch$ **do**

    Neural Network Forward Propagation

    **if** $t_{pre} < t_{post}$,Presynaptic spike early **then**

        Use STBP to update the connection weights between neurons in all layers

        $w \leftarrow w + \Delta w_{stbp}$

        Next use STDP to update the gradient information of the convolutional layer according to Eq. (3)

        $w \leftarrow w + \Delta w_{stdp}$

    **else**

        Same as above, but this time the presynaptic impulse is delayed

        $w \leftarrow w + \Delta w_{stdp} + \Delta w_{stbp}$

    **end if**

**end for**

---

gradient information separately can give full play to the performance of the SNN network and make it have better biological interpretability. For STBP, $\Delta w_{ij}$ represents the change in the connection weight between neuron $i$ and neuron $j$:

$$\frac{dV_j(t)}{dt} = \sum_{i=1}^{N} \Delta w_{ij} D(V(t) - s_j(t) \cdot V_{thr}),  \tag{7}$$

where $D(\cdot)$ represents the neural model of STBP, $V(t)$ represents the membrane voltage of neuron $j$, $s_j$ and $s_j$ represent whether the neuron $i$ and neuron $j$ emit pulses respectively, and $V_{thr}$ is the threshold value of the neuron to emit a pulse. Similarly, for STDP, $P(\cdot)$ represents the neural model of STDP, and it is necessary to consider the pulse emission of the front neuron $i$ and the back neuron $j$ at the same time:

$$\frac{dV_j(t)}{dt} = \sum_{i=1}^{N} \Delta w_{ij} P(V(t) - (s_j(t) + s_i(t)) \cdot V_{thr}).  \tag{8}$$

In the STDP-STBP cyclic gradient update mechanism, we use STBP to update the connection weights between all neurons including the convolutional layers, and then use STDP to update all convolutional layers in the network to implement a cyclic gradient update mechanism.

In the experiment, we also made more attempts, such as using STDP to update the connection weights between neurons in all convolutional layers and fully connected layers in the network and using STBP to update the connection weights between neurons in other layers except convolutional layers and fully connected layers. In this way, STDP and STBP will update the gradients of different layers, reflecting the dynamic combination of neuronal excitability and inhibition.

### 4.3 BRAIN-INSPIRED RECURRENT ITERATIVE LEARNING

In addition to using the STDP-STBP cyclic gradient update mechanism, a certain proportion of STDP and STBP can be added during the training process to update the connection weights between neurons separately. In this way, the excitatory and inhibitory behaviors of neurons can be distinguished during training, and the problem of gradient information not being able to be updated to earlier layers when using STDP can be effectively avoided.

In the human brain, excitatory neurons account for approximately 85%, while inhibitory neurons account for approximately 15%. The fact that excitatory neurons connect to a large number of neurons aligns with the properties of STBP; the fact that inhibitory neurons connect to only a small number of neurons aligns with the properties of STDP. Therefore, in our model, we use STBP to

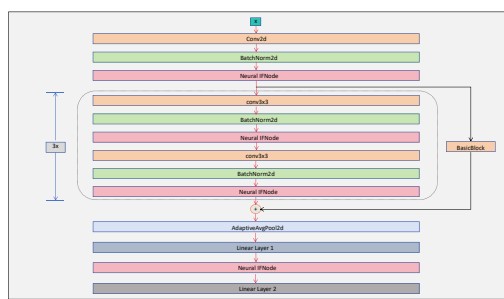

Figure 2: The figure above shows the training model framework, including the backbone network and the residual network, which ensures that the excitation mechanism and the inhibition mechanism can give full play to their respective advantages during training.

represent excitatory neurons and STDP to represent inhibitory neurons. We also use the STDP-STBP cyclic gradient update mechanism to represent excitatory neurons that are closely connected to inhibitory neurons.

In the framework of network model, we use ResNet as the model and build our own network, including convolution layer, normalization layer, activation layer, pooling layer, residual fast and fully connected layer. We use average pooling layers instead of maximum pooling layers to reduce data noise to a certain extent, capture the global information of some features, and make the features smoother. Each residual block contains two sets of convolutional layers, two sets of normalization layers, and two sets of activation layers. For the final output of the model, we expand the single fully connected layer to two sets of fully connected layers plus one set of activation layers, which can effectively extract more advanced features and improve the performance of the model, and can also effectively improve the training effect of STDP.

## 5 EXPERIMENT

### 5.1 DATASETS

We evaluate our approach on a workstation equipped with a 16-core Intel(R) Xeon(R) Platinum 8352V CPU and 4 NVIDIA 4090 GPUs. The network architecture used in the experiment is ResNet, and the experimental datasets are: CIFAR-10 Krizhevsky et al. (2010), CIFAR-100 Lin (2013), MNIST Deng (2012), DVS128 Gesture Hu et al. (2022), and Tiny-imagenet Le & Yang (2015).

CIFAR-10 and CIFAR-100 are classic image classification datasets. They are commonly used for the evaluation and comparison of models in the fields of deep learning and computer vision.The MNIST dataset is one of the most well-known datasets in the field of handwritten digit recognition. DVS128 Gesture is a dynamic vision dataset focused on gesture recognition, captured by a dynamic vision sensor. Due to its address-event representation, DVS128 Gesture is very suitable for SNNs.

### 5.2 EXPERIMENTAL COMPARISON

In this section, we use STDP, STDP-STBP and BIRIL to train our ResNet and compare with other methods. When using STDP training, we use STDP to update the parameters of all convolutional layers in the neural network. STDP-STBP uses STBP to update the parameters of all layers and then uses STDP to update the parameters of the convolutional layer to complete a parameter update loop. BIRIL uses different proportions of STDP, STBP and STDP-STBP for parameter update. In addition, we choose Resnet-18 from STDP-BW-GS Shao et al. (2023) as our base model and draw Figure 3 for comparison. STDP-BW-GS uses STDP to update the parameters of some layers of the neural network, which has certain biological inspiration. Our experimental results are shown in table 1.

Table 1: Comparison of test accuracy of STDP, STDP-STBP and BIRIL with other methods

| Dataset | STDP (Acc. %) | STDP-STBP(ours) (Acc. %) | BIRIL(ours) (Acc. %) | Improvement (Δ Acc. %) | Baseline Methods (Accuracy Gains) |
|---|---|---|---|---|---|
| CIFAR 10 | 42.23 | 93.60 | 93.40 | +2.30 +1.06 +4.46 | • SSTDP: (91.30% → 93.60%) • Diet-SNN: (92.54% → 93.60%) • STDP-BW-GS: (89.14% → 93.60%) |
| CIFAR 100 | 11.54 | 72.25 | 70.89 | +1.63 +0.56 +18.15 | • S-ResNet: (70.62% → 72.25%) • STBP-tdBN: (71.69% → 72.25%) • STDP-BW-GS: (54.10% → 72.25%) |
| MNIST | 89.16 | 99.26 | 99.30 | +0.40 +0.29 +0.06 | • EMSTDP: (98.90% → 99.30%) • BPR: (99.01% → 99.30%) • STDP-BW-GS: (99.24% → 99.30%) |
| DVS128Gesture | 24.31 | 94.80 | 95.49 | +0.89 +4.97 +8.68 | • SNN: (94.60% → 95.49%) • R-STDP: (90.52% → 95.49%) • STDP-BW-GS: (86.81% → 95.49%) |

Note: Improvement (Δ) shows the accuracy gain of STDP-STBP or BIRIL compared with others.

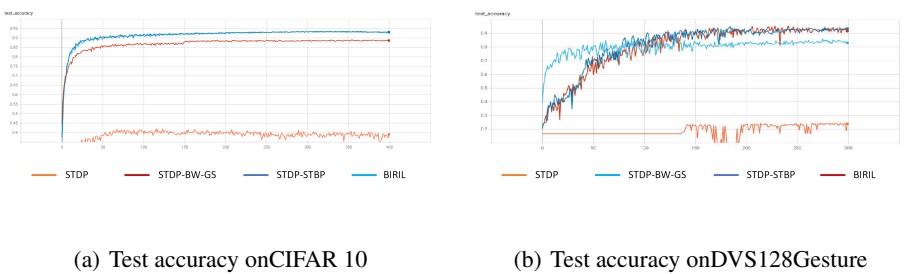

(a) Test accuracy onCIFAR 10       (b) Test accuracy onDVS128Gesture

Figure 3: Performance of test accuracy of STDP, STDP-BW-GS, STDP-STBP and BIRIL

On CIFAR-10, BIRIL achieves an accuracy of 93.40%. STDP-STBP reaches a maximum of 93.60% , which is 2.30% higher than SSTDP Liu et al. (2021), 1.06% higher than Diet-SNN Rathi & Roy (2021), and 4.46% higher than STDP-BW-GS. As shown in fig. 3, STDP-STBP and BIRIL can enhance the expressiveness of neurons and perform well on relatively large datasets such as CIFAR-10.

On CIFAR-100, BIRIL achieves a maximum accuracy of 70.89%. STDP-STBP achieves an accuracy of 72.25% , which is 1.63% higher than S-ResNet Hu et al. (2021), 0.56% higher than STBP-tdBN Zheng et al. (2021), and 18.15% higher than STDP-BW-GS. As shown in fig. 3, experiments show that STDP-STBP and BIRIL have excellent performance on different large datasets and have better biological interpretability.

On MNIST, STDP-STBP achieves an accuracy of 99.26%. BIRIL achieves a maximum of 99.30%, which is 0.40% higher than EMSTDP Shrestha et al. (2021), 0.29% higher than BPR Zhang et al. (2021), and 0.06% higher than STDP-BW-GS. As shown in fig. 3, BIRIL and STDP-STBP have strong model generalization capabilities on the MNIST dataset.

On DVS128 Gesture, STDP-STBP achieves an accuracy of 94.80%. BIRIL achieves a maximum of 95.49% on the test set, which is 0.89% higher than the SNN reported in Amir et al. (2017), 4.97% higher than R-STDP Nadafian et al. (2024), and 8.68% higher than STDP-BW-GS. As shown in fig. 3, this shows that STDP-STBP and BIRIL not only have good performance on static datasets, but also have good generalization ability in processing event-driven data.

Compared with other methods, STDP-STBP alone has an advantage on CIFAR-10 and CIFAR-100, while BIRIL, which combines STDP, STBP, and STDP-STBP, has an advantage on MNIST and DVS128 Gesture. Our STDP-STBP and BIRIL combine the excitation mechanism and inhibition mechanism of neurons, which is more in line with the neuronal structure of the human brain and performs well on different datasets.

Table 2: Experiments on different ratios between STDP, STBP, and STDP-STBP of BIRIL

| Dataset | Ratio of STDP to STBP to STDP-STBP | | | | | Best Ratio |
| --- | --- | --- | --- | --- | --- | --- |
| | 1:4:1(Acc.%) | 1:6:2(Acc.%) | 2:6:1(Acc.%) | 1:8:3(Acc.%) | 3:8:1(Acc.%) | |
| CIFAR 10 | 93.21 | 93.36 | 93.08 | 93.40 | 92.96 | 1:8:3 |
| CIFAR 100 | 70.30 | 70.48 | 69.70 | 70.89 | 69.40 | 1:8:3 |
| MNIST | 99.28 | 99.30 | 99.29 | 99.22 | 99.23 | 1:6:2 |
| DVS128Gesture | 89.93 | 88.89 | 95.49 | 87.50 | 89.58 | 2:6:1 |

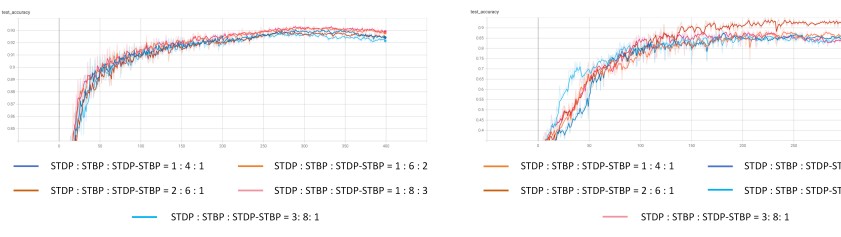

(a) Test accuracy onCIFAR 10        (b) Test accuracy onDVS128Gesture

Figure 4: Performance of test accuracy of different ratios between STDP, STBP, and STDP-STBP

## 5.3 MIXING RATIO TRAINING OF BIRIL

In this section, we examine the impact of different proportions of STDP, STBP, and STDP-STBP cyclic gradient update mechanisms. These different training methods enriched the framework of BIRIL. Through a large number of experiments, we concluded that the optimal ratio of the sum of STDP and STDP-STBP used in the network model to STBP is 1:2. A training framework with this ratio can achieve both excellent performance and good biological interpretability.

As shown in table 2 and Figure 4, we employ a series of ratios (STDP to STBP to STDP-STBP). On CIFAR-10, the highest accuracy is 93.40%, the lowest is 92.96%, and their difference is 0.44%. On CIFAR-100, the highest accuracy is 70.89%, the lowest is 69.40%, and their difference is 1.49%. The best training ratio on both datasets is 1:8:3, indicating that more inhibitory neurons improve performance. On MNIST, the highest accuracy is 99.30%, the lowest is 99.22%, and their difference is only 0.08%, indicating that BIRIL has a wider range of applicability on smaller datasets. On DVS128 Gesture, the highest accuracy is 95.49%, the lowest is 87.50%, and their difference is 7.99%. The best training ratio is 2:6:1. However, increasing the ratio to 3:8:1 reduces accuracy. This shows that a more excited model is more suitable for this dataset, but an overexcited model reduces the model's expressiveness. In addition, if the proportion of STDP or STDP-STBP continues to increase unilaterally on the basis of 1:8:3 or 3:8:1, the network will gradually enter an over-excited or over-inhibited state, which is not conducive to the training results. The different ratios of STDP, STBP, and STDP-STBP make the BIRIL framework both flexible and biologically interpretable. Different ratios can construct BIRIL with different degrees of excitation/inhibition. We can build the most adaptable framework in a targeted manner on different datasets, which greatly enhances the versatility of our model.

## 6 CONCLUSION

By simulating the excitatory and inhibitory mechanisms of neurons in the human brain, our proposed Bidirectional Inhibitory and Regulatory Integrative Learning (BIRIL) model significantly enhances the performance of Spiking Neural Networks (SNNs). Experimental results demonstrate that the BIRIL model exhibits strong expressiveness, superior biological interpretability, and robust generalization capabilities across both small- and large-scale datasets. In future work, we aim to further investigate the dynamic balance between excitation and inhibition in biological neural systems.

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

## A    APPENDIX

### A.1    USE OF LLMS

Large Language Models (LLMs) were used solely to assist with polishing the text.

### A.2    CODE OF ETHICS AND ETHICS STATEMENT

The research conducted in the paper conform, in every respect, with the ICLR Code of Ethics https://iclr.cc/public/CodeOfEthics.

