# OpenReview forum: "Deep Spiking Neural Network with Brain-Inspired Recurrent Iterative Learning"
_ICLR.cc/2026/Conference — ICLR 2026 Conference Withdrawn Submission_

### Official Review · Reviewer_XmiU · 2025-10-26

**Soundness:** 2
**Presentation:** 2
**Contribution:** 2
**Rating:** 6
**Confidence:** 3

**Summary:**

This paper proposes BIRIL, a hybrid training framework for deep spiking neural networks that couples local and global learning under excitation and inhibition dynamics. The method repeatedly combines local plasticity driven by spike trains with global updates driven by error signals, modeling these processes as excitation and inhibition to narrow the gap between biological plausibility and task performance. Experiments on multiple benchmarks show that BIRIL achieves higher test accuracy than representative hybrid and conventional SNN baselines, while improving biological interpretability and training robustness.

**Strengths:**

1.**Novel integration strategy**: The three-cycle training approach combining STDP, STBP, and STDP-STBP in configurable ratios represents an original contribution to hybrid SNN learning. This flexibility allows adaptation to different datasets with varying optimal ratios.

2.**Multi-layer STDP extension**: The paper addresses a known limitation by extending STDP to deep networks through the cyclic gradient update mechanism, enabling parameter updates across multiple layers rather than single-layer applications.

**Weaknesses:**

1. **Unclear algorithmic specification**: The STDP-STBP cyclic gradient update mechanism is poorly defined. Eq. (7)- (8) introduce notation (D(·), P(·)) without clear definitions,and their connection to Algorithm 1 is not established. The claim that “STDP leverages STBP gradients” (Sec. 4.1) lacks a formal update rule showing how STDP uses those gradients.

2. **Weak biological justification**: In Section4.2, the mapping “STDP = inhibitory state” and “STBP = excitatory state”  lacks neuroscientific grounding; inhibition in biology arises from neurotransmitter identity, not distinct learning rules. The biological linkage is metaphorical rather than mechanistic.

3. **Minor issues remain**: the font size in several figures (e.g., Figs. 2–4) is too small and substantially impairs readability. The manuscript also contains grammatical errors and exhibits imprecise, non-rigorous writing.

**Questions:**

See Weakness.

---

### Official Review · Reviewer_JxtM · 2025-10-28

**Soundness:** 1
**Presentation:** 1
**Contribution:** 1
**Rating:** 0
**Confidence:** 5

**Summary:**

This work studies how the local STDP learning rule can be combined with error-backpropagation in the form of STBP. The proposed algorithms computes both updates for a pass, and for the proposed BIRIL algorithm for a proportion of training STDP is applied to inhibitory neurons and STBP to excitatory neurons. The BIRIL approach works just as well as the mixed STBP/STDP approach.

**Strengths:**

I see no strengths in this paper.

**Weaknesses:**

The result is unsurprising as BIRIL essentially performs just as well as STDP-STBP, meaning that there is still an (undetermined) amount of STBP going on in the entire network.

I cannot make out the critical part of the information: compared to just STBP, does adding STDP and/or BIRIL improve learning, such as using less compute, or fewer learning cycles?

STDP as defined is not particularly biologically plausible, and neither are Integrate-and-Fire spiking neurons. See for example the work from Clopath et al, Nature Neuroscience 2010.

the nomenklatura used is non-standard (front neuron, rear neuron -> pre- and postsynaptic neuron).

While STDP-STBP is defined in algorithm 1, the proposed BIRIL algorithm is not formally defined. It seems that for a proportion of training (how is that determined?) STDP is applied to inhibitory neurons and STBP to excitatory neurons.

It is unclear how training is done exactly: what is the stopping criteria, is this the same for all methods?

All figures are illegible when printed.

I really dont understand figure 1, totally unclear.

**Questions:**

Is there any quantifiable benefit from using BIRIL?

**Details Of Ethics Concerns:**

The quality of this submission is so far below what would be expected for a submission to a high-quality technical conference that I would quality this as unprofessional behaviour. This includes illegible graphs and non-existent valid comparisons.

---

### Official Review · Reviewer_vDG3 · 2025-11-01

**Soundness:** 1
**Presentation:** 1
**Contribution:** 1
**Rating:** 0
**Confidence:** 5

**Summary:**

This paper presents a SNN training approach called BIRIL that basically combines biologically inspired STDP and backpropagation (BPTT/STBP).

**Strengths:**

While the general attempt to integrate STDP and BPTT is interesting, I have found many flaws in this paper.

**Weaknesses:**

- The overall approach of this work is pretty problematic and ad-hoc - this is a very ad-hoc mixing of two well known techniques: STDP and BPTT.

- The authors try to combine these two methods in several different variants without providing any fundamental new contributions. In addition, their ways of putting these two methods are not justified and appear to be random. No deep reasoning or analysis of their approach is presented.

- Some of the statements made in the paper are pretty confusing and problematic. It is hard to understand the association of excitatory neurons with BPTT/STBP, and inhibitory neurons with STDP - these are very different things.

- The authors should be clear about whether they compare with the SOTA baseline methods; and demonstration should be also performed on larger datasets.

**Questions:**

- Equations (7, 8) are confusing and problematic. What are $D(\cdot)$ and $P（\cdot)$?

---

### Official Review · Reviewer_xsaT · 2025-11-01

**Soundness:** 1
**Presentation:** 1
**Contribution:** 1
**Rating:** 0
**Confidence:** 4

**Summary:**

This paper introduces a new, more biologically plausible algorithm called BIRIL, which tackles existing methods that overlook biologically realistic mechanisms like spike-timing-dependent computations. However, this algorithm has not been tested on large networks or datasets, so its practical effectiveness remains uncertain. Although this paper states in the abstract that this work paves the way for more energy-efficient AI systems, there is no direct link between biologically plausible, brain-like models and the energy efficiency of real algorithms. Lower energy use is primarily related to hardware, and unfortunately, this paper does not include any hardware analysis.

**Strengths:**

N/A

**Weaknesses:**

I will explain in detail why I believe this paper lacks strengths and, therefore, give it a score of 0. It fails to meet the standards in any aspect.

I list the logical inconsistencies in this paper. (Poor presentation)

1. The abstract states that "conventional SNN training methods often neglect biologically plausible mechanisms such as spike-timing-dependent computations." This paper further mentions in lines 93-95, "The STDP, a biologically inspired mechanism that adjusts synaptic weights based on temporal correlations." So, do existing SNN training algorithms (such as STDP) have biological plausibility? And, do they process temporal information? Moreover, STDP stands for spike-timing-dependent plasticity (as stated in the title of Section 3.3). Why does spike-timing-dependent plasticity neglect spike-timing-dependent computations? Or does this paper believe that STDP is not a training scheme for SNN?
2. The author points out that STBP, like ANN, overlooks the critical role of spike-timing dependent computations. I disagree with this view. First, RNN, as a kind of ANN, including current looped transformers (such as Google's Universal Transformer), has conducted in-depth research on iterative training and inference along the time axis. BPTT specifically discusses how to perform gradient backpropagation over time. I agree that ANN doesn't focus on spikes, but ANN's handling of temporal sequences is already quite advanced. Second, why doesn't STBP address spike-timing dependent computations? This paper also states in lines 104-105 that "STBP effectively captures both spatial and temporal information flow," so can STBP actually process temporal information?

---

I continue to state that the method proposed in this paper is impractical. (Poor contribution)

This paper states, "By co-optimizing biologically plausible dynamics and task-driven performance, our approach unlocks the full potential of SNNs for efficient, hardware-friendly neuromorphic systems," in lines 43-45. It appears that this paper aims to take a step towards resolving the dilemma of SNNs in on-chip learning. However, this paper still uses STBP in the training process. It also fails to demonstrate that STBP can be gradually eliminated under this hybrid learning approach, thus completely transitioning the learning algorithm to STDP, which is more suitable for neuromorphic chip learning. The training method proposed in this paper is merely a rudimentary intermediate product. This paper is more suitable as an undergraduate project.

---

Finally, I believe the experiments in this paper fail to demonstrate that the proposed method is more brain-like, more efficient, or has lower energy consumption. (Poor soundness)

1. There are no experiments on deep SNNs, despite the word "deep" being the first word in the paper's title.
2. Experiments were only conducted on a few toy datasets. Since SEW-ResNet, numerous SNN works have been validated on large datasets such as ImageNet. The experiments in this paper cannot prove the effectiveness of the proposed method.
3. Although the authors emphasize in the abstract and introduction that the proposed method leads to low power consumption, the energy consumption of the design is not discussed in the experiments. Does the SNN trained using the method proposed in this paper have a higher firing rate than usual? A higher firing rate would lead to higher energy consumption.

**Questions:**

I have no doubts about this paper because it is riddled with too many flaws and should be completely rewritten.

---

### Note · Authors · 2025-11-13

I have read and agree with the venue's withdrawal policy on behalf of myself and my co-authors.